# Consumption of Monounsaturated Fatty Acids Is Associated with Improved Cardiometabolic Outcomes in Four African-Origin Populations Spanning the Epidemiologic Transition

**DOI:** 10.3390/nu13072442

**Published:** 2021-07-16

**Authors:** Supal Mehta, Lara Ruth Dugas, Candice Choo-Kang, Pascal Bovet, Terrence Forrester, Kweku Bedu-Addo, Estelle Vicki Lambert, Jacob Plange-Rhule, Walter Riesen, Wolfgang Korte, Amy Luke

**Affiliations:** 1Stritch School of Medicine, Loyola University Chicago, Maywood, IL 60153, USA; smehta1@luc.edu; 2Public Health Sciences, Parkinson School of Health Sciences and Public Health, Loyola University Chicago, Maywood, IL 60153, USA; cchookang@luc.edu (C.C.-K.); aluke@luc.edu (A.L.); 3University Center for Primary Care and Public Health (Unisanté), 1010 Lausanne, Switzerland; pascal.bovet@unisante.ch; 4Unit for Prevention and Control of Cardiovascular Diseases, Ministry of Health, Victoria P.O. Box 52, Seychelles; 5Solutions for Developing Countries, University of the West Indies, Mona, Kingston 7, Jamaica; terrence.forrester@uwimona.edu.jm; 6Department of Physiology, Kwame Nkrumah University of Science and Technology, Kumasi, Ghana; kwekuba@yahoo.com (K.B.-A.); jprhule@gmail.com (J.P.-R.); 7Research Centre for Health through Physical Activity, Lifestyle and Sport, University of Cape Town, Cape Town 7700, South Africa; vicki.lambert@uct.ac.za; 8Center for Laboratory Medicine St. Gallen, 9000 St. Gallen, Switzerland; wf.riesen@bluewin.ch (W.R.); Wolfgang.Korte@zlmsg.ch (W.K.)

**Keywords:** omega 3 fatty acids, omega 6 fatty acids, AA/EPA + DHA ratio, cardiometabolic risk, African-origin, epidemiologic transition

## Abstract

Long-chain omega-3 PUFAs, specifically eicosapentaenoic acid (EPA) and docosahexaenoic acid (DHA), are of increasing interest because of their favorable effect on cardiometabolic risk. This study explores the association between omega 6 and 3 fatty acids intake and cardiometabolic risk in four African-origin populations spanning the epidemiological transition. Data are obtained from a cohort of 2500 adults aged 25–45 enrolled in the Modeling the Epidemiologic Transition Study (METS), from the US, Ghana, Jamaica, and the Seychelles. Dietary intake was measured using two 24 h recalls from the Nutrient Data System for Research (NDSR). The prevalence of cardiometabolic risk was analyzed by comparing the lowest and highest quartile of omega-3 (EPA+ DHA) consumption and by comparing participants who consumed a ratio of arachidonic acid (AA)/EPA + DHA ≤4:1 and >4:1. Data were analyzed using multiple variable logistic regression adjusted for age, gender, activity, calorie intake, alcohol intake, and smoking status. The lowest quartile of EPA + DHA intake is associated with cardiometabolic risk 2.16 (1.45, 3.2), inflammation 1.59 (1.17, 2.16), and obesity 2.06 (1.50, 2.82). Additionally, consuming an AA/EPA + DHA ratio of >4:1 is also associated with cardiometabolic risk 1.80 (1.24, 2.60), inflammation 1.47 (1.06, 2.03), and obesity 1.72 (1.25, 2.39). Our findings corroborate previous research supporting a beneficial role for monounsaturated fatty acids in reducing cardiometabolic risk.

## 1. Introduction

Metabolic syndrome or, more recently, cardiometabolic risk, is captured by a cluster of abnormalities that includes hypertension, central obesity, elevated fasting glucose, and dyslipidemia [1]. These abnormalities are associated with a state of chronic inflammation and an increased risk of developing both type 2 diabetes and cardiovascular disease (CVD) [2]. Globally, the prevalence of this cluster of cardiometabolic abnormalities is increasing at alarming rates [3]. In low-middle income countries (LMICs), the increasing prevalence of both obesity and type 2 diabetes are thought to be attributed to the epidemiological transition. This describes a complex change in disease patterns based on the interactions between demographic, economic and sociologic determinants [4]. Behavioral modifications, such as promoting a healthy diet, and weight reduction have been associated with a reduction in cardiometabolic risk factors [5]. Although weight control is central to addressing the rise in cardiometabolic risk factors, the impact of different dietary components (e.g., fatty acids) remains unclear.

Many studies have focused on the benefits of fiber and mono and polyunsaturated fatty acids in terms of reducing the cardiometabolic risk [6]. To date, however, the majority of studies exploring the relationship between dietary intake and cardiometabolic risk have been conducted in predominantly European-origin populations, with little existing data exploring this relationship in African-origin populations. To our knowledge, there are limited studies that assess how these dietary factors affect health outcomes in countries as they span the epidemiologic transition, as indicated by the human development index (HDI), a country ranking indicating a country’s overall social and economic achievements. Therefore, the objective of this study is to examine the associations between daily consumption of dietary fiber, omega 3 and 6 fatty acids, and monounsaturated fatty acids, and cardiometabolic risk, obesity, and inflammation, in four African-origin populations spanning the epidemiologic transition.

### 1.1. Dietary Fiber Intake

Dietary fiber refers to a group of complex carbohydrates that are not hydrolyzed by digestive enzymes, and thus, are not digested or absorbed in the gut [7]. Dietary fiber has been shown to directly reduce cardiometabolic risk factors such as blood pressure, cholesterol, and levels of inflammatory biomarkers [7,8,9], although the exact mechanism is not known. The two main categories of dietary fiber are soluble fiber and insoluble fiber. Both soluble and insoluble fiber are resistant to digestion; however, soluble fiber is fermented to short-chain fatty acids (SCFA) by colonic bacteria in the large intestine [8]. SCFAs have been shown to suppress proinflammatory mediators, such as TNF-*α* and IL-6 [7]. Insulin resistance is due to the upregulation of these inflammatory markers in insulin-target tissues, such as the liver, adipose tissue, and muscles [10]. Insoluble fiber improves insulin sensitivity by stimulating the secretion of glucagon-like peptide (GLP-1). GLP-1 is an incretin hormone that stimulates postprandial release of insulin and protects beta-cell function [11].

### 1.2. Polyunsaturated Fatty Acids (PUFAs)

There are two main types of polyunsaturated fatty acids (PUFAs): omega-3 and omega-6. Omega-6 fatty acids are largely represented by linoleic acid (LA 18:2 ω-6) and omega-3 fatty acids by alpha-linolenic acid (ALA 18:3 ω-3) [12]. Western diets contain disproportionate levels of omega-6 PUFAs, notably because of increased consumption of corn and other foods that are rich in omega-6 PUFAs. The typical western diet has a ratio of approximately 15/1–16/1, whereas a ratio of 4/1 is recommended for secondary prevention of cardiovascular disease [13]. Diets high in Omega-6 fatty acids are shown to be prothrombotic and proinflammatory, which increases cardiometabolic risk [14], although there is some evidence which indicates that this effect may be muted [15]. A possible mechanism is that the eicosanoid byproducts of omega-6 PUFAs, prostaglandin PGE2 and leukotriene LTB4, are potent mediators of thrombosis and inflammation [16].

Long-chain omega-3 PUFAs, specifically eicosapentaenoic acid (EPA) and docosahexaenoic acid (DHA), are of increasing interest because of their favorable effect on cardiometabolic risk [17]. For example, Stanton et al. conducted a double-blind study which demonstrated that eating foods enriched in omega-3 resulted in a clinically significant reduction in diastolic blood pressure and heart rate [18]. Additionally, studies have found that long-chain omega-3 fatty acids can improve lipid profiles by decreasing serum triglycerides and LDL levels and by increasing HDL levels [19,20]. Omega-3 PUFAs have also been shown to decrease platelet aggregation and thrombus formation [21].

### 1.3. Monounsaturated Fatty Acid Intake

Monounsaturated fatty acids (MUFA) contain a single double bond, as opposed to polyunsaturated fatty acids which contain two or more double bonds [22]. The main type of MUFA found in our diet is oleic acid, which is primarily derived from vegetable oil [23]. Prior randomized controlled trials demonstrate the benefits of a high-MUFA diet compared to a high carbohydrate diet. These studies report improvements in metabolic risk factors such as glycemic control, serum lipids, and blood pressure in both healthy and diabetic patients [24,25,26,27]. Compared to high carbohydrate diet, consumption of MUFAs is associated with decreased glycemic load, which reduces the demand for insulin and increases insulin sensitivity [28]. Additionally, some studies show that oleic acid can reduce blood pressure by improving cell membrane fluidity and by binding to alpha-2-adrenergic receptors [29].

## 2. Methods

### 2.1. Study Population and Ethics Approval

The Modeling the Epidemiologic Transition Study (METS) [25] is a prospective cohort study of 2500 African-origin participants. Participants were recruited from five countries as they span the epidemiologic transition, and include: Ghana, Jamaica, South Africa, the Seychelles and the United States. Using the UN Human Development Index (HDI) 2010, Ghana is a low HDI country, South Africa is a middle HDI country, Seychelles and Jamaica are high HDI countries, and the United States is a very high HDI country [30]. However, for the current analysis, data from South Africa were dropped due to concerns about the quality of the collected dietary data. Baseline measures were obtained between January 2010 and December 2011. A detailed description of recruitment, measures and protocols has been previously published [25]. The protocols were approved by the Institutional Review Board of Loyola University, Chicago, IL, USA (LU#200038); the Committee on Human Research Publication and Ethics of Kwame Nkrumah University of Science and Technology, Kumasi, Ghana; the Research Ethics Committee of the University of Cape Town, South Africa; the Board for Ethics and Clinical Research of the University of Lausanne, Switzerland; and the Ethics Committee of the University of the West Indies, Kingston, Jamaica. Written informed consent was obtained from all participants.

### 2.2. Anthropometry and Biochemical Measures

Body weight (kg) height (cm), and waist circumference (cm) were measured in light clothing and no shoes using standardized equipment across all sites. Body mass index (BMI) was calculated as weight/height^2^ (kg/m^2^), and participants were classified as normal weight (BMI: <25 kg/m^2^), overweight (BMI > 25 kg/m^2^ and <30 kg/m^2^) or obese (BMI > 30 kg/m^2^).

Fasting blood samples were drawn first thing in the morning for analysis of glucose, insulin, lipids, and C-reactive protein (CRP), as previously described [25].

### 2.3. Dietary Intake

A detailed description of the method for capturing dietary intake has previously been published [25]. Briefly, each study participant completed two 24-h recalls using the multiple pass method, separated by 1 week [31,32,33]. Food identification and portion size estimation were approximated from photographs of local foods at each site following methodology developed by the Medical Research Council South Africa [34]. Dietary analysis was performed using the Nutrient Data System for Research (NDSR; University of Minneapolis, MN, USA) [31,32,33]. The primary endpoint measures of interest were total energy intake and macronutrient composition.

### 2.4. Physical Activity Measurement

Physical activity (PA) was assessed using triaxial accelerometers (Actical, Phillips Respironics, Bend, OR, USA) as previously described [35,36]. Briefly, the Actical monitor was worn on the right hip over an 8-day period. Using a SAS macro program [37] and the protocol from the National Center for Health Statistics for the analysis of accelerometry data in the continuous National Health and Nutrition Examination Survey [38]. Daily moderate, and vigorous physical activity minutes were used to estimate total daily minutes of moderate-to-vigorous activity (min/per day).

### 2.5. Clinical Outcomes

Cardiometabolic disease was defined according to the Adult Treatment Panel III criteria [39,40], which stipulate that individuals have at least three of the following cardiometabolic components:Waist circumference > 102 cm in males and >88 cm in females;Elevated blood pressure (≥130/85 mmHg) or receiving treatment;Hypertriglyceridemia (≥150 mg/dL) or receiving treatment;Low high-density lipoprotein (HDL) cholesterol (<40 mg/dL in males and <50 mg/dL in female) or receiving treatment;Elevated fasting plasma glucose (>100 mg/dL) or receiving treatment.

Inflammation was defined as CRP concentrations > 3.0 mg/L [41,42]. Individuals with a BMI ≥ 30 kg/m^2^ were defined as obese [41].

### 2.6. Statistical Analyses

Participant characteristics and cardiometabolic risk factors were summarized using means ± standard deviations (SD), after being examined for normality. Proportions were calculated and presented as *n* (%) for dichotomous variables. Quartiles of total fiber, soluble fiber, and insoluble fiber, and summation of EPA and DHA were determined for each site as well as across all sites based on mean individual dietary intake (g/day). We also calculated the ratio of arachidonic acid (AA) divided by the sum of EPA and DHA [18]. Quartiles of the AA/EPA + DHA ratio were determined for each site as well as across all sites. %Total Energy Consumption (TEC) of monounsaturated acid was determined by subtracting %TEC of fat by %TEC of saturated, polyunsaturated, and trans fatty acids. Quartiles of %TEC of monounsaturated fatty acids were determined for each site as well as across all sites. Comparison of cardiometabolic risks at each site was performed by Pearson’s chi-squared test with statistical significance noted for *p* ≤ 0.05. Multi-variable logistic regression was performed with cardiometabolic outcomes, inflammation, and obesity after adjusting for age, gender, energy intake, smoking, alcohol intake, and PA. Statistical analyses were performed using SAS (version 9.4, Manufacturer, Cary, NC, USA).

## 3. Results

Table 1 presents descriptive characteristics of participants by site. Weight, height, and therefore, BMI, varied greatly across all sites. Ghanaians were shorter (162.6 ± 8.2 cm), lighter (63.4 ± 11.5 kg), and had lower body mass indices (24.1 ± 4.5 kg/m^2^) compared to participants in all the other sites. Conversely, US participants were the tallest (169.9 ± 8.2 cm), heaviest (91.9 ± 24.2 kg) and had the highest body mass indices (31.9 ± 8.4 kg/m^2^). Similarly, Ghanaians had the lowest average waist circumference (81.2 ± 12.0 cm), while US participants were on the highest end (99.6 ± 20.4 cm). Additionally, systolic and diastolic blood pressures were lowest in Ghanaian participants (113.8 ± 14.8 mmHg and 67.1 ± 11.3 mmHg, respectively) and highest in the USA (122.6 ± 16.3 and 80.5 ± 12.7, respectively).

The US participants had the highest mean values of all biochemical measures of cardiometabolic risk factors (total cholesterol, triglycerides, blood glucose, and CRP). The lowest mean values of biochemical measures of cardiometabolic risk were distributed amongst the lower and mid HDI countries. Participants from Ghana had the lowest total cholesterol and blood glucose while participants from the Seychelles had the lowest CRP level. The highest proportion of current and former smokers were from the US (50.6%) and the lowest proportion were from Ghana (4.3%). The US had the highest proportion of participants who consumed alcohol (46.5%) and Ghana had the lowest (13.6%). Not surprisingly, participants from Ghana reported the highest level of daily moderate-to-vigorous PA (34.3 ± 22.6 min/day) while US participants reported least amount of daily activity (23.3 ± 28.9).

Table 2 summarizes habitual dietary intake at the four sites based on the 24-hr dietary recall. The US participants consumed the most calorie-dense diet (2294.5 ± 891.8 kcal/day) compared to the other sites. The US diet was highest in fat (36.6% ± 7.0%), including % saturated fat (11.8% ± 2.9%), high in protein (15.5% ± 4.1%), and low in carbohydrates (45.8 ± 9.4%). Interestingly, participants in the Seychelles had the highest protein intake (18.4 ± 4.7%), reflecting the high quantity of seafood consumed in this island nation. The Ghanaian diet was lowest in fat (21.6% ± 9.1%) and protein (11.9% ± 4.0%), but high in carbohydrates (65.8% ± 10.4%).

Participants in the US consumed the lowest quantity of total dietary fiber (142. ± 7.1 g/day) and insoluble fiber (9.5 ± 5.4 g/day). On the other hand, Ghanaians had the highest intake of total dietary fiber (24.9 ± 9.7) and insoluble fiber (18.8 ± 7.5 g/day). Overall, 43% of Ghanaians met the Institute of Medicine (IOM) fiber guidelines (>14 g fiber/1000 kcal/day), compared to only 9% of Jamaicans, 6% of participants from the Seychelles, and only 3% of participants in the US [42].

US participants also consumed the lowest amount of EPA + DHA (0.15 ± 2.7 g/day) while participants in the Seychelles consumed the highest amount of EPA + DHA (0.69 ± 0.64 g/day). Overall, 52.1% of participants in the Seychelles met the International Society for the Study of Fatty Acids and Lipids (ISSFAL) guidelines (≥0.50 g/day of EPA and DHA) compared to 7.2% of US participants [43]. US participants consumed highest ratio of AA/EPA + DHA (6.5 ± 25.2 g/day) while participants in Ghana consumed the lowest AA/EPA + DHA ratio (0.43 ± 1.4). Overall, 7.2% of US participants consumed a ratio of AA/EPA + DHA < 4:1 versus 17.3% of participants in Ghana. In patients with cardiovascular disease, an AA/EPA + DHA ratio of <4:1 has been associated with a 70% decrease in mortality [14].

US participants consumed the highest %TEC of monounsaturated fatty acids at (16.8 ± 3.7 %TEC) while participants in Ghana consumed the least amount of %TEC of monounsaturated fatty acids (10.0 ± 4.2%TEC). Overall, 69.8% of US participants exceeded the American Heart Association’s guidelines to consume <15% of TEC from MUFA compared to only 8% of participants in Jamaica [44].

### 3.1. Cardiometabolic Risk across the Epidemiologic Transition

Figure 1 compares cardiometabolic risk by site. Participants in the US had the greatest prevalence of cardiometabolic risk factors. Overall, 33% of US participants had 3/5 cardiometabolic risk factors, 41% had inflammation defined as a CRP level > 3, and 52% were obese. Participants in Jamaica had the lowest percentage of participants who met 3/5 cardiometabolic risk factors. However, while the prevalence of cardiometabolic risk was low in Jamaica, 32% of participants presented with inflammation and 31% were obese. Ghanaians had lowest percentage of participants with inflammation (21%) and obesity (9.9%).

### 3.2. Dietary Fiber Analysis

Table 3, the adjusted multivariate logistic regression models demonstrate an inverse relationship between dietary fiber intake and cardiometabolic risk. After adjusting for age, gender, energy intake, physical activity, smoking, and alcohol use, it is established that increasing total fiber, soluble fiber, and insoluble fiber intakes are associated with lower cardiometabolic risk, inflammation, and obesity. Participants who met 3/5 cardiometabolic risk factors were 1.96 times (1.25, 3.06) as likely to be in the lowest quartile of total fiber intake, 1.83 times (1.17, 2.84) as likely to be in the lowest quartile of soluble fiber intake, and 1.80 times (1.16, 2.79) as likely to be in the lowest quartile of insoluble fiber intake compared to participants in the highest quartile of fiber intake. Additionally, participants with elevated CRP were 1.70 times (1.28, 2.43) as likely to be in the lowest quartile of total fiber intake and 1.46 times (1.03, 2.07) as likely to be in the lowest quartile of insoluble fiber. Finally, participants in the lowest quartile of total, soluble, and insoluble fiber were more likely to be obese; however, these associations were no longer statistically significant after controlling for site.

### 3.3. Omega-3 and Omega-6 Fatty Acid Analysis

The prevalence of cardiometabolic risk was analyzed by chi-square analysis comparing the lowest quartile of EPA+ DHA (Q1) with the highest quartile (Q4), and by comparing participants who consumed an AA/EPA + DHA ratio of ≤4:1 and >4:1. The trend for the lower prevalence of cardiometabolic risk factors with increased intake of EPA and DHA and a total AA/EPA + DHA ratio of ≤4:1 was seen for cardiometabolic conditions in some sites. However, it should be noted that differences in the prevalence of cardiometabolic risk factors and EPA and DHA intake and an AA/EPA + DHA ratio ≤4:1 and >4:1 were modest, with statistical significance achieved only in the prevalence of increased cardiometabolic risk in the Seychelles based on an AA/EPA + DHA ratio of ≤4:1 and >4:1.

In Table 4, the adjusted multivariate logistic regression models demonstrate an inverse relationship between EPA + DHA intake and cardiometabolic risk. After adjusting for age, gender, energy intake, physical activity, smoking, and alcohol, increasing EPA + DHA intake is associated with protection from higher cardiometabolic risk (2.16 (1.45, 3.20)), inflammation (1.59 (1.17, 2.16)), and obesity (2.06 (1.50, 2.82)). However, these associations are no longer seen after controlling for site. The adjusted multivariate logistic regression models demonstrate how an AA/EPA + DHA ratio of ≤4:1 also protects against increased cardiometabolic risk (1.80 (1.24, 2.60)), inflammation (1.47 (1.06, 2.03)), and obesity (1.72 (1.25, 2.39)). However, these associations are no longer seen after controlling for site.

### 3.4. The Association of Monounsaturated Fatty Acids and Cardiometabolic Risk across the Epidemiological Transition

The prevalence of cardiometabolic risk factors was analyzed by chi-square analysis comparing the lowest quartile of %TEC monounsaturated fatty acids (Q1) with the highest quartile (Q4), and by comparing participants who consumed <15% TEC of monounsaturated fatty acids and ≥15% TEC. The trend for the lower prevalence of cardiometabolic risk with increased %TEC of monounsaturated fatty acid intake and ≥15% TEC of monounsaturated fatty acids was seen at some sites.

In Table 5, the adjusted multivariate logistic regression models demonstrates no significant association between higher cardiometabolic risk, obesity, and inflammation when comparing quartiles of %TEC of MUFA or when comparing a diet with %TEC of MUFA ≥ 15% or <15%.

## 4. Discussion

In our study, we found a heterogeneous pattern of cardiometabolic risk factors in the four populations spanning the epidemiologic transition. Despite this, the prevalence of cardiometabolic risk factors is lowest in Ghana (lowest HDI) and highest in the US (highest HDI) compared to the other study. The data also suggest that LMICs may be experiencing a rise in obesity rates and nutrition-related noncommunicable diseases by adopting a more western-style diet. Indeed, as populations become more affluent, both the total calories consumed as well as the macronutrient component changes, with the consumption of inexpensive ultra-processed foods. A systemic evaluation conducted by the Bill and Melinda Gates foundation found that a suboptimal diet is responsible for more deaths globally than other risk factors, including tobacco smoking [45]. While significant emphasis has been placed on promoting diets that are low in sodium, sugar, and fat, the assessment shows that diets low in whole grains, fruits, nuts and seeds, vegetables, and omega-3 fatty acids account for more than 2% of global deaths [46].

### 4.1. Dietary Fiber Intake

When comparing the prevalence of cardiometabolic risk between participants consuming either the lowest or highest daily total, soluble, and insoluble dietary fiber across all sites, a significant association is seen for higher cardiometabolic risk, inflammation (indicated by elevated CRP levels), and obesity in the participants with the lowest total, soluble, and insoluble fiber intake. The only exception is the association between inflammation and soluble fiber intake. This study corroborates previous studies that demonstrate how dietary fiber can be protective against cardiometabolic risk [7,8,9]. Similarly to fiber, meat and saturated fats are not completely digested and are fermented in the gut to produce short chain fatty acids. However, the metabolism of meats and saturated fats produces inflammatory and pro-neoplastic nitrogenous metabolites, such as nitrosamines, phenolics, and p-cresol [47]. However, high fiber diets can counteract these inflammatory and neoplastic pathways [48]. The Institute of Medicine (IOM) recommends consuming at least 14 g of fiber per day, and the US Department of Agriculture (USDA) recommends that women consume at least 22 g of fiber per day and that men consume 38 g/d [42]. While 42.2% of participants in Ghana met the 14 g/day cut-off recommendation by the IOM, less than 10% of participants in the Seychelles, Jamaica, and the United States met these guidelines.

### 4.2. Omega-3 and Omega-6 Fatty Acid Intake

A significant association is seen between higher cardiometabolic risk, inflammation (indicated by elevated CRP levels), and obesity in the participants consuming the lowest quantity of EPA and DHA versus participants in the highest quartile. Our study supports previous studies that demonstrate the beneficial effects of EPA and DHA in reducing the risk of cardiometabolic outcomes [18,19,20,21]. Unfortunately, in most countries and regions, the intake of omega-3 fatty acids is substandard [49]. ISSFAL recommends consuming ≥0.50 g/day of EPA and DHA [50], while the American Heart Association (AHA) recommends that patients with coronary artery disease consume 1 g of EPA and DHA per day and that patients with hypertriglyceridemia consume 2–4 g of EPA and DHA per day [51]. In the Seychelles, 52.2% of participants met these guidelines. However, across the other sites, most participants did not meet the ISSFAL guidelines. In fact, only 7.2% of participants in the United States consumed the recommended amount of EPAH and DHA.

Across all sites, a significant association is seen between cardiometabolic risk, inflammation, and obesity in the participants consuming an AA/EPA + DHA ratio of >4:1. Only 0.7% of US participants consumed a diet with an AA/EPA + DHA ratio < 4:1. In Ghana, 17.2% of participants consumed a diet with an AA/EPA + DHA ratio < 4:1. The increased omega-6 fatty acid consumption in western diets can be attributed to high intake of processed foods and the increased use of corn or sunflower oil (which are rich in omega-6 fatty acids) [52]. Additionally, animal feed is now predominantly grain instead of grass in many countries. This has caused an increase in the level of arachidonic acid, a type of omega-6 fatty acid, in meat, eggs, and dairy products [53].

### 4.3. Monounsaturated Fatty Acid Intake

When comparing the prevalence of cardiometabolic risk between participants consuming either the lowest and highest %TEC of monounsaturated fatty acids across all sites, no significant association was seen with cardiometabolic risk, inflammation, and obesity. Similarly, no significant association was seen with cardiometabolic risk, inflammation, and obesity when comparing participants who consumed %TEC of monounsaturated fatty acids greater than or less than 15%. Current studies investigating the relationship between MUFA intake and cardiometabolic risk report inconsistent findings [54,55]. A possible explanation for the discrepancy is that the source of MUFA is often not specified. In Western diets, MUFA is mainly derived from animal-based sources, such as dairy and meat. Animal-based sources of MUFA are higher in saturated fats compared to plant-based sources, such as olive oil, nuts, or avocado, which could be confounding the results [56]. While plant-based sources of MUFA have been associated with health benefits, health organizations such as the United States Department of Agriculture and the American Diabetes Association do not provide specific dietary recommendations for MUFA intake for healthy individuals or for individuals with chronic conditions. The current MUFA recommendations vary between 12% and 25% of TEC, equaling a remarkable range of ~30–70 g/day for a 2500 kcal diet [57].

### 4.4. Significance of Site

After we controlled our analysis for country of origin, the association between dietary factors and cardiometabolic risk was no longer significant. This could be attributed to the disproportionate number of participants in the US who have cardiometabolic risk (33.3%) compared to <10% of the population at the other three sites. Overall, the prevalence of metabolic risk factors, such as waist circumference and cholesterol levels, is much lower in Ghana (lowest HDI) compared to the US (highest HDI), while sites with a medium HDI fall between those observed in Ghana and the United States. This underlies large differences in other risk factors that are highly related to mean BMI across the included countries.

Despite these trends, the overall quality of diet across all of the sites remains poor. The epidemiologic transition has caused a shift in dietary patterns resulting in an increase in the quantity and a decline in the quality of macronutrients, mainly fats and carbohydrates. In many of these countries, traditional foods that are rich in whole grains and dietary fiber are being replaced by highly refined carbohydrates [57]. In addition, there has been an increase in the consumption of refined sugars and sugar-sweetened beverages, along with animal products [58].

Popkin et al. conducted a study to analyze trends in obesity and waist circumference across low-and-middle income countries, from 1990 to 2010, among adults aged 19–49. The study found an increase in BMI at each percentile in most countries, particularly at the 95th percentile. For example, in 1993, the BMI cut-off for the 95th percentile for women aged 30 years was 28.7. However, in 2008, the BMI cut-off for the 95th percentile was 31.3 [59]. In addition to an increase in BMI cut-offs, the study found an overall increase in waist circumference at each BMI cut-off. Many studies show an association between waist circumference and an increased cardiometabolic risk [60]. The nutrition transition has resulted in dietary changes that can stray away from traditional healthful diets and towards a pattern of highly processed and refined foods. The data demonstrate the need to develop policy at a global level to improve education and access to nutritious foods, and to promote the avoidance of over nutrition (particularly sugar beverages), in order to slow down the high rates of obesity and the tremendous societal health and economic costs associated with it.

### 4.5. Limitations

This study represents one of the few studies that examines cardiometabolic risk across multiple international sites spanning the epidemiological transition in people of African origin. This was achieved using central training of research staff, standardized questionnaires, protocols, analysis and methodology to allow us to conduct these comparisons across all sites. However, we recognize that there are limitations to our study. Firstly, dietary information was dependent on two self-reported 24-h recalls. Twenty-four-hour recalls frequently result in an underestimation of the portion size and do not account for variations in diet across many days and seasons [61]. Additionally, how we quantified dietary data differs from other studies. For example, studies that looked into the association between omega-6 and omega-3 fatty acids and cardiometabolic risk use the omega-3 index as the primary outcome. The omega-3 index is calculated as the sum of EPA and DHA expressed as a percentage of the total fatty acids in erythrocyte membranes [18]. The omega-3 index correlates with the level of EPA and DHA in cardiac tissue [62], and is a predictor of cardiovascular events and all-cause mortality [19,63,64]. Lastly, within each site, the sample size was relatively small, and not necessarily representative of that country as a whole. The study has a fairly narrow age range and does not include individuals over the age of 50. While this allows for direct comparison across sites, individuals older than 50 are more likely to have cardiometabolic disease and, generally, higher levels of cardiovascular risk factors.

## 5. Conclusions

Diet has become one of the most important risk factors to reduce the global burden of noncommunicable chronic diseases. The purpose of our study is to further elucidate our understanding of which dietary nutrients are commonly, or even universally, considered important. Given the likely benefits of fiber and omega-3 fatty acids shown in this study, further concerted effort, through multi-sectoral interventions and policy, should be made to improve availability and access to foods that are rich in these nutrients and to adjust food-based guidelines so that they are specific to a given country or region.

## Figures and Tables

**Figure 1 nutrients-13-02442-f001:**
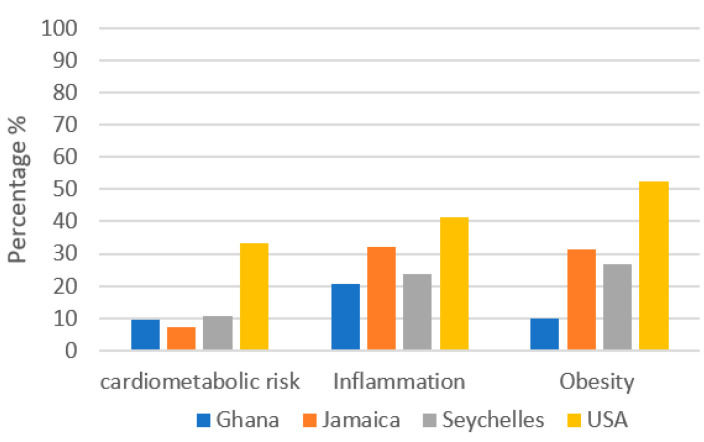
Comparison of prevalence for participants with at least 3/5 cardiometabolic risk factors by site.

**Table 1 nutrients-13-02442-t001:** Descriptive characteristics of study population by site.

	Ghana (*n* = 487)	Jamaica (*n* = 398)	Seychelles (*n* = 484)	United States (*n* = 444)
Demographics:				
Female (*n*, %)	287 (58.9)	244 (61.3)	261 (53.9)	227 (51.2)
Age	34.2 ± 6.7	34.4 ± 6.1	36.2 ± 5.6	35.3 ± 5.6
Anthropometrics				
Weight (kg)	63.4 ± 11.5	75.9 ± 17.1	75.9 ± 17.0	92.2 ± 24.3
Height (cm)	162.6 ± 8.2	168.1 ± 8.9	167.2 ± 8.8	169.9 ± 9.0
BMI (kg/m^2^)	24.1 ± 4.5	27.0 ± 6.5	27.1 ± 5.6	32.0 ± 8.5
Waist circumference (cm)	81.2 ± 12.0	87.1 ± 14.1	88.7 ± 12.0	99.8 ± 20.4
Systolic blood pressure (mmHg)	113.8 ± 14.8	118.4 ± 14.2	116.2 ± 14.7	122.6 ± 16.3
Diastolic blood pressure (mmHg)	67.1 ± 11.3	71.7 ± 11.3	73.0 ± 10.7	80.5 ± 12.7
Biochemical Measures				
Cholesterol (mg/dL)	161.3 ± 35.3	162.8 ± 33.9	170.9 ± 35.4	181.0 ± 38.4
HDL-C (mg/dL)	46.1 ± 14.4	46.5 ± 12.1	47.7 ± 12.8	50.8 ± 14.6
Triglycerides (mg/dL)	81.8 ± 40.2	73.2 ± 36.4	79.8 ± 60.6	97.5 ± 57.5
Blood glucose (mg/dL)	100.3 ± 12.3	92.9 ± 9.3	100.6 ± 29.2	103.2 ± 32.9
C-reactive peptide (mg/dL)	4.7 ± 13.3	4.2 ± 6.3	3.1 ± 4.5	6.0 ± 11.0
Lifestyle habits				
Smoker or ex-smoker (*n*, %)	21 (4.3)	130 (32.7)	106 (21.9)	224 (50.6)
Consumes alcohol (*n*, %)	66 (13.6)	160 (40.2)	204 (42.2)	206 (46.5)
Moderate to vigorous PA (min/day)	34.3 ± 22.6	23.2 ± 19.3	28.9 ± 20.9	23.3 ± 28.9

**Table 2 nutrients-13-02442-t002:** Dietary analysis of study population by site.

	Ghana (*n* = 487)	Jamaica (*n* = 398)	Seychelles (*n* = 484)	United States (*n* = 444)	*p*-Value
Energy (kcal)	1848.8 ± 496.3	1893 ± 582.8	1843.6 ± 593.9	2294.5 ± 891.8	<0.0001
% Energy from fat	21.6 ± 9.1	25.7 ± 6.6	28.4 ± 7.7	36.6 ± 7.0	<0.0001
% Energy from saturated fat	7.1 ± 4.1	9.5 ± 4.0	8.4 ± 2.9	11.8 ± 2.9	<0.0001
% Energy from monounsaturated fat	8.2 ± 3.7	8.5 ± 2.7	9.0 ± 3.0	13.6 ± 3.1	<0.0001
% Energy from polyunsaturated fat	4.6 ± 2.5	5.4 ± 2.1	8.6 ± 3.4	7.9 ± 2.8	<0.0001
% Energy from carbohydrates	65.8 ± 10.4	58.6 ± 8.4	51.3 ± 9.4	45.8 ± 9.4	<0.0001
% Energy from protein	11.9 ± 4.0	14.6 ± 3.9	18.4 ± 4.7	15.5 ± 4.1	<0.0001
Dietary fiber	24.9 ± 9.7	15.9 ± 8.3	13.6 ± 7.2	14.2 ± 7.1	<0.0001
Soluble fiber (g)	6.0 ± 2.8	4.7 ± 2.6	3.9 ± 2.2	4.6 ± 2.4	<0.0001
Insoluble fiber (g)	18.8 ± 7.5	11.2 ± 6.1	9.6 ± 5.4	9.5 ± 5.4	<0.0001
Meeting 14 g fiber/1000 kcal (*n*, %)	207 (42.5)	35 (8.8)	28 (5.8)	14 (3.2)	<0.0001
Omega 3:					
Average EPA + DHA g/day	0.65 ± 0.81	0.30 ± 0.40	0.69 ± 0.64	0.15 ± 2.7	<0.0001
EPA + DHA > 0.5 g/day (*n*, %)	193 (39.6)	76 (19.1)	252 (52.1)	32 (7.2)	<0.0001
AA/EPA + DHA ratio					
Average AA/EPA + DHA ratio	10.0 ± 9.5	8.3 ± 3.1	1.0 ± 2.7	6.5 ± 25.2	<0.0001
AA/EPA + DHA ratio < 4:1 (*n*, %)	85 (17.3)	32 (6.4)	31 (6.3)	3 (0.7)	<0.0001
Monounsaturated fatty acids:					
%TEC	10.0 ± 4.2	10.8 ± 3.1	11.4 ± 3.6	16.8 ± 3.7	<0.0001
>15% TEC (*n*, %)	66 (13.5)	32 (8.0)	76 (15.7)	309 (69.8)	<0.0001

**Table 3 nutrients-13-02442-t003:** Adjusted odds ratios for cardiometabolic risk based on quartiles of total fiber, soluble fiber, and insoluble fiber intake in four countries across the epidemiological transition.

**Total Fiber Analysis**						
	**Not Controlled for Site**			**Controlled for Site**		
**Quartiles of Total Fiber**	**3/5 CM Risk Factors**	**Inflammation**	**Obesity**	**3/5 CM Risk Factors**	**Inflammation**	**Obesity**
0.0–10.34 g (Q1)	1.96 (1.25, 3.06) *	1.70 (1.18, 2.43) *	2.76 (1.90, 4.02) *	1.43 (0.85, 2.42)	1.27 (0.84, 1.93)	1.05 (0.68, 1.62)
10.34–15.05 g (Q2)	1.60 (1.05, 2.42) *	1.67 (1.20, 2.32) *	2.52 (1.78, 3.57) *	1.40 (0.86, 2.25)	1.37 (0.95, 1.99)	1.23 (0.82, 1.83)
15.05–22.03 (Q3)	0.97 (0.63, 1.49)	1.21 (.87, 1.67)	1.62 (1.14, 2.29) *	.86 (0.54, 1.36)	1.07 (0.76, 1.51)	1.01 (0.69, 1.48)
>22.03(Q4)	Reference	Reference	Reference	Reference	Reference	Reference
**Soluble Fiber Analysis**						
**Quartiles of Soluble Fiber**	**3/5 CM Risk Factors**	**Inflammation**	**Obesity**	**3/5 CM Risk Factors**	**Inflammation**	**Obesity**
0.0–3.00 g (Q1)	1.83 (1.17, 2.84) *	1.25 (0.88, 1.80)	1.85 (1.28, 2.67) *	1.59 (0.98, 2.56)	1.08 (0.74, 1.58)	1.13 (0.75, 1.67)
3.00–4.34 g (Q2)	1.37 (0.90, 2.08)	1.56 (1.12, 2.16)	1.78 (1.26, 2.51)	1.19 (0.76, 1.87)	1.37 (0.97, 1.93)	1.15 (0.79, 1.67)
4.34–6.00 (Q3)	1.16 (0.77, 1.76)	1.08 (0.78, 1.49)	1.55 (1.11, 2.16)	1.05 (0.68, 1.61)	1.01 (0.72, 1.40)	1.28 (0.90, 1.84)
>6.00 (Q4)	Reference	Reference	Reference	Reference	Reference	Reference
**Insoluble Fiber Analysis**						
**Quartiles of Insoluble Fiber**	**3/5 CM Risk Factors**	**Inflammation**	**Obesity**	**3/5 CM risk Factors**	**Inflammation**	**Obesity**
0.0–6.99 g (Q1)	1.80 (1.16, 2.79) *	1.46 (1.03, 2.07) *	2.68 (1.86, 3.86) *	1.25 (0.74, 2.09)	1.05 (0.70, 1.57)	1.01 (0.66, 1.55)
6.99–10.57 g (Q2)	1.52 (1.00, 2.30) *	1.59 (1.15, 2.20) *	2.32 (1.65, 3.28) *	1.24 (0.77, 1.99)	1.23 (0.85, 1.78)	1.06 (0.71, 1.58)
10.57–16.29 (Q3)	1.01 (0.65, 1.56)	1.04 (0.75, 1.45)	1.54 (1.08, 2.18) *	0.95 (0.60, 1.51)	0.91 (0.64, 1.29)	1.00 (0.67, 1.46)
>16.29 (Q4)	Reference	Reference	Reference	Reference	Reference	Reference

Multivariate model adjusted for age, gender, energy intake, physical activity, alcohol intake, and smoking. * *p* ≤ 0.05. CM = cardiometabolic.

**Table 4 nutrients-13-02442-t004:** Adjusted odds ratios for cardiometabolic risk based on the AA/EPA + DHA ratio of 4:1 intake in four countries across the epidemiological transition.

	**Not Controlled for Site**			**Controlled for Site**		
**AA/EPA + DHA 4:1**	**3/5 CM Risk Factors**	**Inflammation**	**Obesity**	**3/5 CM Risk Factors**	**Inflammation**	**Obesity**
≤4:1	Reference	Reference	Reference	Reference	Reference	Reference
>4:1	1.80(1.24, 2.60) *	1.47 (1.06, 2.03) *	1.72 (1.25, 2.39) *	1.03 (0.69, 1.55)	1.04 (0.74, 1.47)	0.85 (0.60, 1.22)
**Quartiles of EPA + DHA**	**3/5 CM Risk Factors**	**Inflammation**	**Obesity**	**3/5 CM Risk Factors**	**Inflammation**	**Obesity**
0.0–0.07 g (Q1)	2.16(1.45, 3.20) *	1.59 (1.17, 2.16) *	2.06(1.50, 2.82) *	1.05 (0.65, 1.70)	0.98 (0.69, 1.41)	0.80 (0.54, 1.18)
0.07–0.23 g (Q2)	1.30 (0.85, 1.97)	1.08 (0.78, 1.49)	1.20 (0.87, 1.67)	0.83 (0.52, 1.34)	0.82 (0.58, 1.15)	0.72 (0.49, 1.04)
0.23–0.58 (Q3)	1.23 (0.80, 1.87)	0.83 (0.60, 1.15)	1.04 (0.75, 1.44)	1.19 (0.77, 1.85)	0.74 (0.53, 1.03)	0.85 (0.60, 1.21)
>0.58 (Q4)	Reference	Reference	Reference	Reference	Reference	Reference

Multivariate model adjusted for age, gender, energy intake, physical activity, alcohol intake, and smoking. * *p*≤
0.05. CM = cardiometabolic.

**Table 5 nutrients-13-02442-t005:** Adjusted odds ratios for cardiometabolic risk based on %TEC monounsaturated fatty acids <15% and >15% in four countries across the epidemiological transition.

	**Not Controlled for Site**			**Controlled for Site**		
**% of MUFA Intake**	**3/5 CM Risk Factors**	**Inflammation**	**Obesity**	**3/5 CM Risk Factors**	**Inflammation**	**Obesity**
<15%	0.52 (0.38, 0.70)	0.72 (0.57, 0.91)	0.50 (0.39, 0.65)	1.25 (0.86, 1.82)	0.97 (0.72, 1.31)	1.12 (0.81, 1.54)
≥15%	Reference	Reference	Reference	Reference	Reference	Reference
	**Not Controlled for Site**			**Controlled for Site**		
**Quartile of %TEC MUFA Intake**	**3/5 CM Risk Factors**	**Inflammation**	**Obesity**	**3/5 CM Risk Factors**	**Inflammation**	**Obesity**
0.0–8.9	0.39 (0.26, 0.60)	0.51 (0.36, 0.71)	0.31 (0.21, 0.45)	1.08 (0.64, 1.81)	0.80 (0.54, 1.19)	0.94 (0.61, 1.45)
8.9–11.8	0.42 (0.28, 0.63)	0.70 (0.51, 0.95)	0.47 (0.34, 0.65)	1.16 (0.72, 1.88)	1.04 (0.72, 1.50)	1.13 (0.76, 1.68)
11.8–15.30	0.60 (0.42, 0.86)	0.74 (0.55, 1.00)	0.82 (0.61, 1.11)	1.15 (0.77, 1.72)	0.97 (0.70, 1.36)	1.54 (1.08, 2.20)
>15.3	Reference	Reference	Reference	Reference	Reference	Reference

## Data Availability

The data presented in this study are available on request from the corresponding author.

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
