# Peer review of "Consumption of Monounsaturated Fatty Acids Is Associated with Improved Cardiometabolic Outcomes in Four African-Origin Populations Spanning the Epidemiologic Transition"

_nutrients, 2021, doi:10.3390/nu13072442_

Round 1
Reviewer 1 Report
This manuscript reports that both low intake of EPA+DHA and consuming an omega 6:3 ratio >4:1 are associated with increased cardiometabolic risk, inflammation and obesity in 4 African-origin populations based on dietary recall analysis. Although the findings are interesting and significant, I have several major concerns about the manuscript as follows.
- My main concern is the fatty acid data, which are the focus and key content of this study. The data are highly unreliable, as they were derived from only two self-reported 24-hour recalls of 4 different populations with very different dietary patterns and food sources. Unlike other macronutrients, PUFA content in foods can vary greatly under such circumstances. Fatty acid profile of blood samples from the participants is much more meaningful and reliable. The authors should measure RBC fatty acid composition of sampled subjects.
- As presented in the manuscript, the objective of this study and results cover dietary fiber, omega-3 and omega-6 fatty acids and monounsaturated fatty acids, why didn’t the abstract and title of the manuscript reflect all the findings (didn’t mention the fiber and MUFA results)?
- Given the very high heterogeneity of the 4 populations studied, another analysis of potential correlations between quartiles of fatty acid intake and cardiometabolic outcomes within each/individual site may be necessary.
- The terms: “Omega-3 Index” and “Omega-6:3 ratio” were misused in the manuscript (Abstract & Table 2). The food content of EPA and DHA is NOT matched with Omega-3 Index. The ratio of AA/EPA+DHA doesn’t reflect the omega-6/omega-3 ratio.
- As the association between omega-6:3 ratio and cardiometabolic outcomes is very significant and a major finding of the study, omega-6:3 ratio should be a key word of this manuscript and should be emphasized in the discussion and conclusions (increased intake of omega-6 PUFA may be a risk factor).
- CRP is the only one parameter for inflammation in this study. Blood TNF-a and IL-6 are the more sensitive and important inflammatory parameters to be measured .
Minor point:
Reference 16 seems incorrect (please check the author and title).
Author Response
Dear Reviewer 1.
Thank you for your helpful comments. We have addressed each in a point-by-point manner.
1. My main concern is the fatty acid data, which are the focus and key content of this study. The data are highly unreliable, as they were derived from only two self-reported 24-hour recalls of 4 different populations with very different dietary patterns and food sources. Unlike other macronutrients, PUFA content in foods can vary greatly under such circumstances. Fatty acid profile of blood samples from the participants is much more meaningful and reliable. The authors should measure RBC fatty acid composition of sampled subjects.
Thank you for this comment. We agree that self-report 24hr dietary recalls may provide unreliable data, in fact we have published the precise level of under-reporting in our data set, using the doubly labeled water: Orcholski et al. Br J Nutr . 2015 Feb 14;113(3):464-72. doi: 10.1017/S000711451400405X. Epub 2015 Jan 13. Furthermore, while RBC fatty acid composition would provide a more accurate estimation of fatty acid levels, the data collection in the 2,500 participants was completed in 2009-2011, and samples were analyzed in 2013. However, we are currently completing a new wave, and given the reviewer’s suggestion, will include RBC fatty acid analysis.
2. As presented in the manuscript, the objective of this study and results cover dietary fiber, omega-3 and omega-6 fatty acids and monounsaturated fatty acids, why didn’t the abstract and title of the manuscript reflect all the findings (didn’t mention the fiber and MUFA results)?
Our results indicate that the consumption of PUFAs, not MUFAs is associated with improved CM outcomes. We have adjusted the title to reflect this. The new title is: “Consumption of mono unsaturated fatty acids is associated with improved cardiometabolic outcomes in 4 African-origin populations spanning the epidemiologic transition.”
3. Given the very high heterogeneity of the 4 populations studied, another analysis of potential correlations between quartiles of fatty acid intake and cardiometabolic outcomes within each/individual site may be necessary.
We agree that there is a high level of heterogeneity between the 4 populations, and as such included the country of origin as a dummy variable in the multiple linear regression analysis. As can be noted from our results, this resulted in differences no longer being statistically significant, after accounting for country of origin.
4. The terms: “Omega-3 Index” and “Omega-6:3 ratio” were misused in the manuscript (Abstract & Table 2). The food content of EPA and DHA is NOT matched with Omega-3 Index. The ratio of AA/EPA+DHA doesn’t reflect the omega-6/omega-3 ratio.
Thank you for highlighting the incorrect use of “Omega-3 Index” and the omega-6/omega-3 ratio. We have corrected these in the abstract and Table 2, and throughout the manuscript.
5. As the association between omega-6:3 ratio and cardiometabolic outcomes is very significant and a major finding of the study, omega-6:3 ratio should be a key word of this manuscript and should be emphasized in the discussion and conclusions (increased intake of omega-6 PUFA may be a risk factor).
Thank you for this suggestion, we have added the key words.
6. CRP is the only one parameter for inflammation in this study. Blood TNF-a and IL-6 are the more sensitive and important inflammatory parameters to be measured .
We agree that CRP represents only one measure of inflammation, and will be measuring TNF-a and IL-6 are the future follow-up visits in the cohort.
7. Minor point: Reference 16 seems incorrect (please check the author and title).
We have corrected reference 16.

Reviewer 2 Report
I red with interest the manuscript by Mehta and colleagues, investigating the impact of dietary omega 3 fatty acids on cardiometabolic risk.
My major concern is that conclusions derived from data collected through the 24HR record that, as reported by the authors in the limitation of the study, frequently result in an underestimation of the portion size and do not account for variations in diet across many days and seasons.
Minor comments:
- I suggest to include in the analysis, if available, data on the presence of instrumental steatosis, type 2 diabetes or glucose intolerance and metabolic syndrome, all factors that increase cardiovascular risk.
Author Response
Dear Reviewer 2.
1. My major concern is that conclusions derived from data collected through the 24HR record that, as reported by the authors in the limitation of the study, frequently result in an underestimation of the portion size and do not account for variations in diet across many days and seasons.
We agree that dietary intake data collected through 24hr recall may be prone to under-reporting. In fact, we have published the exact level of under-reporting in this cohort using the doubly-labeled water method: Orcholski et al. Br J Nutr . 2015 Feb 14;113(3):464-72. doi: 10.1017/S000711451400405X. Epub 2015 Jan 13. We have noted this as a significant study limitation in the limitation section.
2. Minor comments: I suggest to include in the analysis, if available, data on the presence of instrumental steatosis, type 2 diabetes or glucose intolerance and metabolic syndrome, all factors that increase cardiovascular risk.
Thank you for this suggestion. Unfortunately, we do not have data for instrumental steatosis, nor glucose intolerance. We included the type 2 diabetes and metabolic syndrome data in the cardiometabolic risk score in the original manuscript.
